# Effect of Methionine Supplementation on Serum Metabolism and the Rumen Bacterial Community of Sika Deer (*Cervus nippon*)

**DOI:** 10.3390/ani12151950

**Published:** 2022-07-31

**Authors:** Yan Wu, Xiaolan Guo, Dehui Zhao, Chao Xu, Haoran Sun, Qianlong Yang, Qianqian Wei, Huazhe Si, Kaiying Wang, Tietao Zhang

**Affiliations:** 1Jilin Key Laboratory for Molecular Biology of Special Economic Animals, Institute of Special Animal and Plant Sciences, Chinese Academy of Agricultural Sciences, Changchun 130112, China; 15797861336@139.com (Y.W.); guoxiaolan0511@126.com (X.G.); zhaodehui1129@163.com (D.Z.); xuchao@caas.cn (C.X.); solomoncat@163.com (H.S.); qianlongy@126.com (Q.Y.); weiqian8823@163.com (Q.W.); 2College of Animal Science and Technology, Hebei Normal University of Science and Technology, Qinhuangdao 066004, China; 3College of Animal Science and Technology, Jilin Agricultural University, Changchun 130118, China; sihuazhe1989@163.com; 4Jilin Special Animal Feeding and Comprehensive Utilization of Science and Technology Innovation Center, Institute of Special Animal and Plant Sciences, Chinese Academy of Agricultural Sciences, Changchun 130112, China

**Keywords:** glutathione peroxidase, methionine, rumen bacteria, ruminal fermentation, sika deer

## Abstract

**Simple Summary:**

During the antler-growing period, sika deer (*Cervus nippon*) need a large amount of protein from feed for rapid antler growth. Antlers are the most important economic product of sika deer. Methionine is the first or second limiting amino acid in the diets of sika deer, which greatly limits the effective utilization of protein in the feed, while methionine has biological functions such as antioxidant and immune function improvement. In this study, we evaluated for the first time the effect of methionine supplementation in the diet on antler-growing sika deer and tried to investigate the changes in rumen microorganisms and obtain better production efficiency through methionine regulation. The results showed that supplementation with appropriate methionine improved the antioxidant and immune function of the sika deer, while obtaining better antler weight.

**Abstract:**

Methionine is the first or second limiting amino acid for ruminants, such as sika deer, and has a variety of biological functions such as antioxidant activity, immune response, and protein synthesis. This study aimed to investigate the effects of methionine supplementation on antler growth, serum biochemistry, rumen fermentation, and the bacterial community of sika deer during the antler-growing period. Twelve 4-year-old male sika deer were randomly assigned to three dietary groups supplemented with 0 g/day (*n* = 4, CON), 4.0 g/day (*n* = 4, LMet), and 6.0 g/day (*n* = 4, HMet) methionine. No significant difference (*p* > 0.05) was found in the production performance between the three groups, but antler weight was higher in both the LMet and HMet groups than in the CON group. Methionine supplementation significantly increased the serum glutathione peroxidase activity (*p* < 0.05). The serum immunoglobulin G level was significantly higher in the HMet group than in the other two groups (*p* < 0.05). No significant effect was found on the apparent amino acid digestibility of the three groups, but cysteine and methionine digestibility were higher in the LMet group. The serum hydroxylysine level was significantly lower in the LMet and HMet groups, whereas the serum lysine level was significantly lower in the HMet group compared with the CON group (*p* < 0.05). The LMet group had the highest but a nonsignificant total volatile fatty acid content and significantly higher microbial protein content in the rumen than the CON group (*p* < 0.05). The phyla Bacteroidetes, Firmicutes, and Proteobacteria were dominant in the rumen of the sika deer. The principal coordinate analysis (PCoA) and analysis of similarities (ANOSIM) results showed a significant change in the bacterial composition of the three groups (*p* < 0.05). The relative abundance of *Prevotella* and *Rikenellaceae*-RC9 was significantly higher in the LMet group compared with the CON group and CON and HMet groups, respectively. These results revealed that methionine supplementation improved the antioxidant activity and immune status, affecting amino acid metabolism and rumen microbial composition of the sika deer.

## 1. Introduction

As an extremely important species category in livestock farming (including beef cattle, dairy cattle, sheep, etc.), the popular intensive farming model has improved the efficiency of land use, but also brings a large amount of greenhouse gases generated by feed processing, intestinal emissions, and manure disposal compared to free range grazing. In addition, the cultivation of large amounts of genetically modified feed ingredients also poses a threat to the ecosystem, which has many negative impacts on the environment and increases the hidden costs of farming [1,2]. Existing research hopes to mitigate the environmental hazards associated with animal farming by actively seeking diversified organic feed ingredients and improving the efficiency of feed utilization [3]. Determining more precise methionine levels in diets can improve the efficiency of nutrient use in animals, which can effectively reduce resource waste and greenhouse gas emissions.

Methionine is an essential amino acid directly involved in protein synthesis. It influences many physiological functions of the body [4,5]. Methionine supplementation in the diet can significantly improve animal growth and production performance [6]. However, too high or too low levels of methionine often have adverse effects [7,8]. Methionine is often the first or second limiting amino acid in the common maize–soybean meal type of ruminant diets [9,10]. The appropriate levels of methionine should be investigated in the diets of different animals at different stages of production.

Methionine is involved in or regulates various biochemical reactions in animals. Studies have shown that methionine can improve animal production performance by alleviating oxidative damage [8]. Methionine, the ultimate donor of methyl groups required for DNA methylation, promotes protein deposition by regulating the expression of factors such as myostatin, the myogenin gene, the myocyte enhancer factor-2 family, and insulin-like growth factor-1 (IGF-1) [11,12,13]. The sika deer is an economically important animal in the ruminant suborder, as it provides high-quality meat and produces the valuable medicinal antler herb [14]. In China, it is widely believed that the two-branch antler has a higher use value than the three-branch antler. The market price of the two-branch antler is higher, and harvesting them twice has become a common production method to obtain a higher economic return [15]. A number of factors influence the growth process of the antler. Huang et al. confirmed that supplementing methionine in the diet could improve the growth performance of fawns of sika deer by increasing their body weight [16]. Stubbs et al. identified methionine as a critical amino acid in regulating IGF-1 expression in the sheep liver [17]. IGF-1, which is indirectly regulated by methionine, is thought to be an essential factor in antler growth [18]. Therefore, we speculated that methionine supplementation may affect antler growth. Methionine supplementation also increases serum methionine levels in Holstein cows and improves milk protein production [6]. These results suggested that methionine supplementation in the diet might benefit the productive performance of sika deer.

As the sika deer has a compound stomach, its rumen is home to a large number of diverse microorganisms, which play an irreplaceable role in the digestion and absorption of nutrients in the feed [19]. Microorganisms can break down nutrients such as cellulose and protein in feed and convert them into volatile fatty acids that are more favorable for animals to absorb and use, such as acetate and propionate, which can stimulate the growth of rumen epithelium and affect nutrient absorption [20]. The microbial proteins synthesized provide nearly half of the protein source to sika deer [21]. In addition, different microbial compositions in the rumen can also affect the utilization of nutrients by animals, altering protein and lipid deposition and further affecting animal growth [22]. Bickhart et al. concluded that ruminal microorganisms influence milk protein and milk fat content in the milk of Holstein cows and that specific microbiomes improve the quality of dairy products [23]. Previous studies showed that methionine supplementation altered the microbial composition and affected fermentation in the rumen [24]. Thus far, the effects of methionine supplementation on the microbial composition and fermentation in the rumen in antler-bearing deer are still unclear. This study aimed to investigate the impacts of methionine supplementation on the production performance, serum biochemistry, rumen fermentation, and microbiological composition of sika deer during antler growth.

## 2. Materials and Methods

### 2.1. Experimental Design

Twelve 4-year-old male sika deer (mean body weight = 103.2 ± 16.7 kg) were used in this study. These sika deer had a similar time point of hard antler button casting around 19 June 2021. The experimental site is located in Shuangyang District, Jilin Province, China (longitude: 125.724, dimension: 43.539, temperature: 19–28 °C, wind scale: <3). All animal procedures were approved and authorized by the Animal Ethics Committee of the Chinese Academy of Agricultural Sciences.

The 12 deer were randomly assigned to 3 groups, with 4 animals in each group. Each deer was maintained in an individual pen. The animals were fed a diet based on corn silage and concentrate (35:65, dry matter basis), and randomly assigned to one of the three experimental diets (Table 1): a basal diet with 0 g/day (CON), 4.0 g/day (LMet), or 6.0 g/day (HMet) arginine. The animals were fed twice each day at 4:00 am and 5:00 pm (a total of 2.8 kg diet (dry matter)) and had free access to drinking water. The experiments were conducted for 5 weeks after the hard antler button caste, with 1 week for adaptation followed by 4 weeks of dietary treatments.

### 2.2. Sample Collection

For three consecutive days before the end of the trial, a partial collection method was used to collect manure samples each day before the morning feed. Four fresh manure samples of approximately 100 g each were collected from each pen. Hair and grit were removed from the samples, and they were sprayed with 10% dilute sulfuric acid to fix nitrogen, dried to a constant weight at 65°C, and stored for use.

At the end of the experiment, the animals were anesthetized using an anesthetic gun (Chlorpromazine Hydrochloride Injection, 2.0 mL/100 kg body weight) before the morning feed. The antlers were cut with a sterilized saw and weighed. A blood sample of 10 mL was collected by jugular venipuncture, and the serum was obtained by centrifugation at 4000× *g* for 10 min at 4 °C. Ruminal fluid (approximately 200 mL) was obtained via the rumen in the morning before feeding. The first 100 mL of the rumen fluid was discarded to avoid saliva contamination. The samples were transferred to liquid nitrogen and then stored at −80 °C for further analysis. For the collection and processing of these samples we refer to the method of Si H [25].

### 2.3. Measurement of Serum Biochemical Properties

The concentration of serum triglycerides (TGs), total cholesterol, high-density lipoprotein cholesterol (HDL-C), low-density lipoprotein cholesterol (LDL-C), glucose, total protein (TP), albumin (ALB), globulin (GLB), alkaline phosphatase (ALP), and aspartate aminotransferase (AST) were analyzed using commercial colorimetric kits (Nanjing Jiancheng Bioengineering Institute, Nanjing, Jiangsu, China) using a Beckman AU480 automatic biochemistry analyzer (Vitalab Selectra E, Spankeren, The Netherlands). Serum concentrations of immunoglobulin A (IgA), immunoglobulin M (IgM), and immunoglobulin G (IgG) were quantified using enzyme-linked immunoassay kits (MLBIO, Shanghai, China). The serum antioxidant levels were measured using kits in strict accordance with the kit instructions (Nanjing Jiancheng Bioengineering Institute, Nanjing, Jiangsu, China). The total superoxide dismutase (T-SOD) activity was measured by the hydroxylamine method, glutathione peroxidase (GSH-PX) activity by the colorimetric method, catalase (CAT) activity by the ammonium molybdate method, and total antioxidant capacity (T-AOC) by the 2,2’-Azinobis-(3-ethylbenzthiazoline-6-sulphonate) method. Method and reagent information is available through the website (http://www.njjcbio.com/, accessed on 15 October 2021).

The feed and manure samples were determined by the oxidative acid hydrolysis method concerning GB/T 15399-2018 for cysteine and methionine and by the acid hydrolysis method concerning GB 5009.124-2016 for lysine and threonine using a Hitachi L-8900 fully automatic amino acid analyzer (Hitachi Technology, Tokyo, Japan). The apparent digestibility of amino acids (%) = 100 – (100 × *A*/*A*_1_ × *B*_1_/*B*), where *A* is the percentage of 2 mol/L hydrochloric acid insoluble ash in the diet, *A*_1_ is the percentage of 2 mol/L hydrochloric acid insoluble ash in the feces, *B*_1_ is the percentage of amino acid in the feces, and *B* is the percentage of amino acid in the diet [26]. The samples of serum were diluted with trichloroacetic acid and then centrifuged for 5 min at 10,000× *g* to precipitate protein, and the supernatant was used directly after centrifugation. The concentration of serum amino acid was quantified by ion-exchange chromatography (Hitachi L-8900 amino acid analyzer) [16].

### 2.4. Determination of Rumen Fermentation Parameters

To determine the volatile fatty acids in the rumen fluid, the rumen was centrifuged at 5400 rpm for 10 min at 4 °C, and the supernatant was added at 5:1 (supernatant:internal standard) to a 25% metaphosphoric acid solution containing 2-ethylbutyric acid of internal standard, mixed well, and frozen at −20 °C overnight. Again, the rumen was centrifuged at 10,000 rpm for 10 min at 4 °C, and the supernatant was evaluated using a gas chromatograph (GC-6800), which was configured with a Φ 6 mm × 2 m quartz glass-filled column (stationary phase 15% free fatty acid polyester, stretcher 80–100 mesh Chromosorb) [27]. The ammoniacal nitrogen concentration in the rumen fluid was determined using the alkaline sodium hypochlorite–phenol spectrophotometric method, and the standard T/TAIA 0004-2020 was used for reference. Microbial proteins in the rumen fluid were determined using the kits from Nanjing Jiancheng Technology Co. A PHS-3C pH meter was used to determine the pH of the rumen fluid [28].

### 2.5. DNA Extraction, Amplification, Sequencing, and Bioinformatics Analysis

The DNA was extracted using the cetyltrimethylammonium bromide method [29], followed by agarose gel electrophoresis to check the purity and concentration of DNA. The sample was diluted to 1 ng/µL using sterile water in a centrifuge tube. Phusion High-Fidelity Polymerase Chain Reaction (PCR) Master Mix with GC Buffer (New England Biolabs, Ipswich, MA, USA) and high-performance, high-fidelity enzymes were used for PCR to ensure amplification efficiency and accuracy. The PCR product was detected by electrophoresis using a 2% agarose gel. The PCR product that passed the test was purified by magnetic beads, quantified by enzyme labeling, mixed in equal amounts according to the concentration of the PCR product, incorporated thoroughly, and then detected by electrophoresis using a 2% agarose gel, and the target bands were recovered using a recovery kit provided by Qiagen (Redwood City, CA, USA). Libraries were constructed using a TruSeq DNA PCR-Free Sample Preparation Kit (Illumina, San Diego, CA, USA). The libraries were quantified by Qubit and Real-time Quantitative Polymerase Chain Reaction (Q-PCR) and then sequenced using NovaSeq6000 [19,30,31,32].

The Uparse algorithm (Uparse v7.0.1001, http://www.drive5.com/uparse/ (accessed on 15 October 2021)) was used to cluster all the effective tags of all samples, and the sequences were clustered by default parameters with 97% consistency (identity) into operational taxonomic units (OTUs). A representative sequence of OTUs was selected based on the principle of its algorithm, and the sequence with the highest frequency of occurrence in OTUs was selected as the representative sequence of OTUs. Chao1, Shannon, Simpson, and ACE indices and UniFrac distances were calculated using Qiime software (version 1.9.1). The intergroup variance analysis of the alpha diversity index was performed using R software. The PCoA plots were plotted using R software (version 2.15.3). The PCoA analyses were performed using the weighted correlation network analysis (WGCNA), stats, and ggplot2 packages of R software. The adonis analyses were performed using the adonis function of the R vegan package.

### 2.6. Statistical Analysis

The one-way analysis of variance (ANOVA) was used to test the statistical significance of the alpha diversity index, antler growth performance, serum biochemical parameters, apparent digestibility of amino acids, serum amino acids, rumen fermentation parameters, and relative abundance of bacteria among the three groups. When ANOVA tests indicated significant differences between means, differences between groups were determined using the Duncan test. The analysis process uses the analysis software SPSS (IBM SPSS Statistics 26; IBM-SPSS Inc., Chicago, IL, USA).

## 3. Results

### 3.1. Antler Weight, Immune Levels, and Antioxidant Capacity of the Three Groups

As shown in Table 2, methionine supplementation increased the weight and the mean daily weight gain of antlers. The serum IgG levels were significantly higher in the HMet group than in the other two groups (*p* < 0.05), and the highest IgA levels were found in the LMet group. However, no significant difference was present. In addition, the serum GSH-PX activity was significantly higher in the LMet group than in the other two groups, and significantly higher in the HMet group than in the CON group (*p* < 0.05).

### 3.2. Comparison of Serum Biochemical Parameters among Different Groups

As shown in Table 3, methionine supplementation had no significant effect on the serum biochemical parameters of the three groups. However, the LMet group had higher GLB levels than the other two groups. Compared with the CON group, methionine supplementation reduced the serum ALP activity in deer, with the lowest reduction occurring in the HMet group.

### 3.3. Effect of Methionine Supplementation on Apparent Amino Acid Digestibility and Serum Amino Acids

The results showed that methionine supplementation had no significant effect on the apparent digestibility of amino acids in sika deer and did not affect the efficiency of amino acid absorption. On the contrary, the apparent digestibility of Cys and Met was higher in the LMet group (Table 4). Table 5 shows that methionine supplementation significantly reduced serum Lys and serum Hylys concentrations in sika deer (*p* < 0.05). 

### 3.4. Effect of Methionine Supplementation on Rumen Fermentation in Sika Deer

Table 6 shows that methionine supplementation did not have a significant effect on the volatile fatty acids in the rumen of the three groups of sika deer. However, the total volatile fatty acid content increased in the LMet group, and the range of different types of volatile fatty acids changed. In addition, the microbial protein concentration of the rumen fluid was significantly higher in the LMet group than in the CON group (*p* < 0.05). The pH of the rumen fluid increased with the level of methionine supplementation, but the changes were not significant.

### 3.5. Bacterial Community Composition in the Three Groups

An average of 84,552 valid sequences were obtained per sample in this experiment, and sequences were clustered into OTUs with 97% agreement, yielding a total of 2394 OTUs after removing the data results for archaea, unknown, and no blast hit, which were annotated in the Silva138 database (http://www.arb-silva.de/ (accessed on 15 October 2021)) [33] as a basis for analysis.

At the phylum level, the top 10 bacteria in relative abundance were identified, and the results indicated that Bacteroidota (CON = 33.1 ± 4.2%, LMet = 49.8 ± 6.1%, HMet = 37.2 ± 6.0%), Firmicutes (CON = 43.8 ± 6.0%, 34.2 ± 5.6%, HMet = 39.0 ± 8.6%), and Proteobacteria (CON = 12.2 ± 9.1%, LMet = 4.3 ± 1.6%, HMet = 14.0 ± 10.0%) were the most abundant bacteria in the rumen fluid, with the LMet group having a significantly higher abundance of Bacteroidota than the other two groups (*p* < 0.05). At the genus level, *Prevotella* had the highest abundance (CON = 10.4 ± 3.2%, LMet = 16.0 ± 2.5%, HMet = 14.2 ± 2.3%), accounting for more than 10%. *Pseudomonas* (CON = 7.8 ± 6.6%, LMet = 1.6 ± 0.5%, HMet = 8.3 ± 8.1%), *Rikenellaceae_*RC9 (CON = 7.4 ± 0.6%, LMet = 11.5 ± 1.5%, HMet = 6.4 ± 2.6%), *Christensenellaceae_*R-7 (CON = 5.1 ± 2.3%, LMet = 3.5 ± 1.3%, HMet = 3.6 ± 0.9%), and *Ruminococcus* (CON = 6.6 ± 1.2%, LMet = 5.0 ± 1.3%, HMet = 6.4 ± 1.3%) were also widespread in the rumen fluid of the three groups (Figure 1). The results of the alpha diversity analysis (Figure 2) showed that the differences in the OTU number, Chao1, ACE, and Shannon index of the rumen fluid flora of the three groups of sika deer were not significant (*p* > 0.05).

The PCoA results and adonis analysis showed (Figure 3) that the colony composition of the LMet group was significantly separated compared with that of the CON and HMet groups based on the unweighted UniFrac distance, Bray–Curtis distance, and weighted UniFrac distance matrices (adonis: *p* < 0.05 (CON vs. LMet, LMet vs. HMet)), while the colony composition of the CON and HMet groups was not significantly separated (adonis: *p* > 0.05 (CON vs. HMet)) (Table 7). By comparing the differences in the relative abundance of bacteria at the genus level, the relative abundance of *Prevotella* was found to be significantly higher in the LMet group than in the CON group, and the relative abundance of *Rikenellaceae_*RC9 was substantially higher than that in the CON and HMet groups (*p* < 0.05). In addition, a decreasing trend was observed in the abundance of NK4A214 in the LMet and HMet groups compared with the CON group (*p* < 0.1). Meanwhile, the relative abundance of unidentified*_Oscillospiraceae_*UCG-005 and UCG-002 was also a trend in the relative abundance of the LMet group compared with the CON group.

## 4. Discussion

This experiment showed that methionine supplementation was beneficial to the productive performance of sika deer (Table 2). The present experimental conditions suggested that antler production positively correlated with the amount of methionine added. Still, as sika deer are a valuable animal resource in China, the limited number of experimental animals available for this study might have been influenced by this factor, resulting in nonsignificant differences between the three groups. Methionine supplementation affected serum immune levels, with a significant increase in the serum IgG level in the HMet group and an increase in the IgM level (Table 2). Previous studies demonstrated that lymphocytes have a higher requirement for methionine, because in lymphocytes precursors such as choline and homocysteine are not met by methionine regeneration through the methionine cycle [34,35]. Wang Han et al. also demonstrated that methionine supplementation increased the plasma IgG level in dairy cows [36]. Therefore, methionine levels should be more than the growth maintenance requirements of animals to meet the needs of the immune system. Furthermore, methionine supplementation enhanced the antioxidant capacity of sika deer and significantly upregulated the GSH-PX activity (Table 2), a result we speculated might be attributed to an increase in the substrate glutathione (GSH) levels. Methionine can generate homocysteine via the trans-sulphuration pathway, which is converted to cystathionine by cystathionine-β-synthase, serine, and vitamin B6. Cystathionine is cleaved to produce cysteine and α-butyric acid, and cysteine is catalyzed by enzymes to form γ-glutamylcysteine, which is converted to GSH by glutathione synthase and glycine [37,38]. GSH, as a substrate, further reduces hydrogen peroxide to water, catalyzed by GSH-PX, to complete the scavenging of oxygen radicals [39]. In line with these results, Zhou Z et al. found that methionine supplementation significantly improved the antioxidant capacity and immune levels of dairy cows and resulted in better production performance. Taken together, these results suggested that methionine supplementation could improve the immune and antioxidant status of sika deer and achieve good production performance [40,41,42]. 

Serum biochemical indicators reflect the absorption and metabolism of nutrients by the animal organism [43]. In this study, methionine supplementation did not have a significant effect on these substances, but the globulin content increased in both the LMet and HMet groups (Table 3), which was consistent with the previous trend in immunoglobulins. ALP is widely distributed in a variety of tissues and organs in animals. Studies have shown that ALP is considered a marker of bone calcification and its activity reflects the functional status of osteoblasts, as well as that ALP promotes the deposition of calcium ions on collagen, which eventually forms bone tissue [44,45,46]. The growth of the antlers of sika deer is a gradual calcification process, and as the calcification increases, the collagen content of the antlers gradually decreases and their medicinal value is severely reduced [14,47]. The results of this study showed that although no statistically significant differences were found in the ALP activity due to insufficient experimental animals, its activity decreased with methionine supplementation levels, which may be beneficial in slowing down the calcification process in antlers. Studies have shown that methionine is an important amino acid for collagen synthesis [48], so supplementation with methionine might be beneficial for collagen synthesis during antler growth.

The experimental results (Table 4) showed no statistical difference in the effect of methionine supplementation on the apparent digestibility of amino acids in the sika deer, but the LMet group showed an increase in cysteine and methionine digestibility. In agreement with the results of this study, Ylinen V et al. found that methionine supplementation significantly increased the digestibility of cysteine and methionine [49]. Notably, the serum amino acid results showed (Table 5) that methionine supplementation in the diet of sika deer reduced the serum lysine and hydroxylysine levels in the absence of significant changes in the apparent digestibility of lysine, suggesting that the bioavailability of these two amino acids may have been enhanced. This might be related to serum lysine and hydroxylysine deamidation to promote collagen synthesis. This result verified the conjecture that methionine supplementation downregulates the ALP activity, affecting the collagen content in antlers. Previous studies showed that lysine was also one of the most important limiting amino acids in ruminants, with a variety of biological functions [50], and that lysine and hydroxylysine form the cross-linked bonds of collagen molecules through the ε-NH2 oxidative deamination reaction [51,52], of which antlers contained more than 25% [53]. Similarly, Liu and Inhuber et al. [54,55] found that methionine supplementation resulted in a decrease in serum lysine and better production performance. This explains the improved antler weight and daily gain in the LMet and HMet groups. However, many differences still exist in the physicochemical properties of the large number of amino acids in the organism that interact with each other to influence the metabolism and utilization processes. The specific ways in which methionine affects the utilization of lysine, hydroxylysine, or other amino acids need further in-depth investigation.

Ruminal microorganisms can produce short-chain fatty acids by fermenting carbohydrates such as cellulose and hemicellulose, which are the primary source of energy for ruminants, achieving more than 50% of their energy requirements [56]. Previous studies have shown that Bacteroidota and Firmicutes are the dominant phylum in the deer family [25,31,57], and they are also the most central ruminal bacteria with the most significant percentage of abundance in other ruminants, such as dairy cows [58]. This was consistent with the experimental results of this study (Figure 1). Bacteroidota have been reported to be significantly associated with cellulose degradation [59] and can play an essential function in cellulose degradation in sika deer whose diet consists mainly of roughages such as maize straw and twigs and leaves. Jami Elie et al. found that Firmicutes are important bacteria for cellulose degradation and volatile fatty acid production in cattle [60]. In addition, they also play an essential role in the body’s defense against attack by foreign pathogens [61]. At the genus level, *Prevotella*, the dominant bacterium with the highest abundance, was involved in the degradation of proteins and carbohydrates [25]. Jiang et al. concluded that *Christensenellaceae_*R-7 played a vital function in the metabolism of amino acids and lipids [61]. In a study on the fermentation of cellulose and wheat straw by *Ruminococcus*, it was found that it produced significantly higher levels of acetate and propionate than other metabolites [62]. Zhou et al. found that *Rikenellaceae_*RC9 was essential for improving lipid metabolism in the intestine of mice. In addition, *Pseudomonas* was also found to be one of the dominant bacteria in the rumen of plum deer, which was not found in other deer species [21,25,61,63,64]. The reason may be the differences in the living environment and dietary conditions of the animals. *Pseudomonas* has been reported to be powerful for protein fermentation, but it is relatively poor at fermenting carbohydrates [65]. The experimental results showed that supplementation with methionine does not affect the production of volatile fatty acids by microorganisms (Table 6). This suggested that methionine supplementation under the present experimental conditions had little effect on the fermentative capacity of the rumen microorganisms, but the ratio of acetic acid to propionic acid changed, and we speculated that the fermentation pattern in the rumen may have changed [66]. PCoA analysis and adonis analysis based on unweighted UniFrac distance, Bray–Curtis distance, and weighted UniFrac distance matrices showed significant differences in the Bacteroidota structure among the three groups of sika deer (Figure 3, Table 7). At the phylum level, the LMet group had significantly higher levels of Bacteroidota, which might enhance the degradation of cellulose. At the genus level, the LMet group also showed a significant increase in *Prevotella* and *Rikenellaceae_*RC9 (Figure 4). Studies have shown that *Prevotella* was able to generate short-chain fatty acids, mainly acetic acid, through the degradation of carbohydrates [67], and that high levels of *Prevotella* enhanced the fermentation of complex polysaccharides and improved glucose metabolism [68], which explained the rise in acetic acid production in the LMet group. In addition, Balakrishnan B et al. suggested that *Prevotella* maintained the stability of the immune system by enhancing butyrate production [67], which was consistent with the results of this study. Furthermore, Zhao et al. suggested that *Rikenellaceae_*RC9 positively correlated with the production of total volatile fatty acids and acetic acid, while affecting the pH of rumen fluid [69]. Ruminal fluid pH showed an increasing trend in this study, but the pH values were stable between 6.5 and 7 in all three groups, suggesting that supplementation with methionine did not affect rumen health [70]. This study showed that changes in the abundance of these two microorganisms did alter the volatile fatty acid production in the rumen. On the contrary, the changes in the HMet group were not significant (*p* > 0.05) relative to the CON group (Figure 4). Similarly, in the rumen microbiome of sika deer fed diets supplemented with different levels of arginine, the effect of high-dose addition on the abundance of *Bacteroids* spp. was found to be much less variable than in the low-dose group [25]. This might be related to the rate of degradation of different concentrations of amino acids in the rumen [71]. Microbial protein is an essential source of directly available protein for ruminants [72]. In this study, methionine supplementation had a significant effect on the microbial protein content in rumen (Table 6), with the highest levels of microbial protein in the LMet group. This suggested that supplementation with appropriate methionine could improve the utilization of nitrogen by ruminal microorganisms. The results for serum ammonia showed the highest concentration in the HMet group, suggesting that it is also possible that the higher concentration of methionine degraded to a greater extent, resulting in a lower utilization of methionine by microorganisms in the HMet group than in the LMet group [73,74,75]. These results suggested that methionine supplementation in the diet altered the bacterial composition in the rumen of sika deer, thereby affecting the fermentation patterns in the rumen.

## 5. Conclusions

The results of this study showed that methionine supplementation in the diet improved antioxidant and immune levels and increased antler weight in sika deer during the antler growth period. At the same time, there was no negative impact on the utilization of amino acids. In addition, methionine supplementation in the diet affected the microbial composition and altered fermentation in the rumen. At present, we need to further understand the apparent digestibility of other nutrients to determine the efficiency of feed utilization, and to analyze the differences in antler composition among groups and the expression of antler-growth-related genes to investigate the molecular mechanism of methionine regulation of nutrient deposition. Overall, methionine supplementation can improve animal health in response to possible production-related stress, which provides more animal welfare and more economic benefits to the farmer.

## Figures and Tables

**Figure 1 animals-12-01950-f001:**
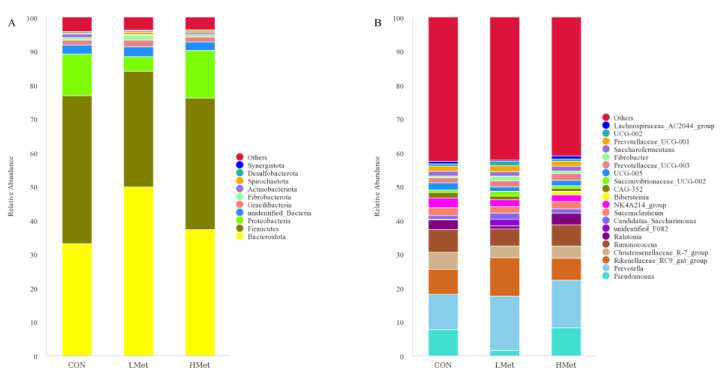
Bacterial composition in the rumen of sika deer at the phylum (**A**) and genus (**B**) levels.

**Figure 2 animals-12-01950-f002:**
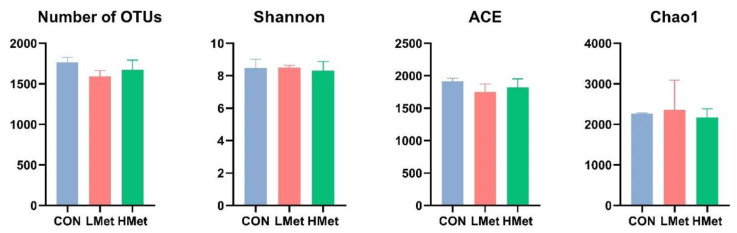
Comparisons of the alpha diversity of the bacteria in the rumen of sika deer.

**Figure 3 animals-12-01950-f003:**
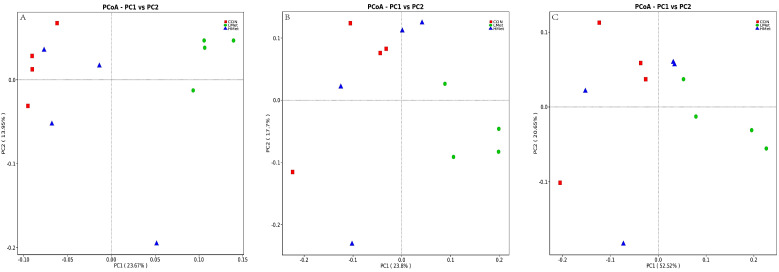
Comparisons of the bacterial communities in the rumen of sika deer. Principal coordinate analyses based on unweighted UniFrac distances (**A**), Bray–Curtis distance (**B**), and weighted UniFrac distance (**C**).

**Figure 4 animals-12-01950-f004:**
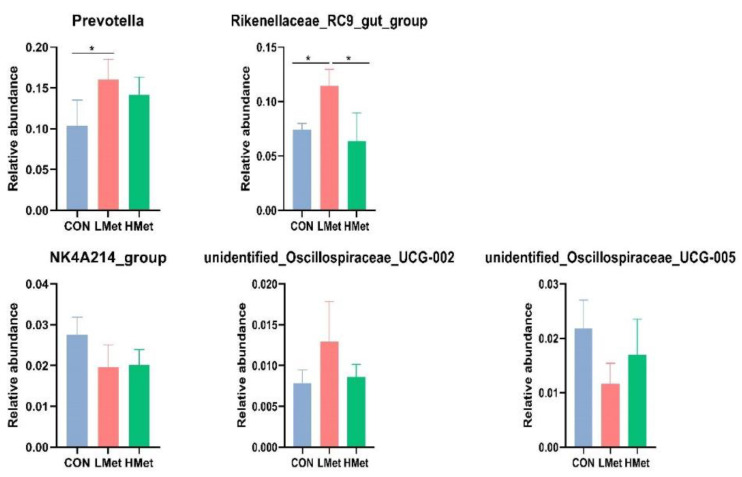
Bacterial analysis in genus-level differences in sika deer (* indicates *p* < 0.05).

**Table 1 animals-12-01950-t001:** Basic dietary composition and nutrient levels (dry matter basis).

Ingredients	(g/100 g)	Nutrient Levels	(%)
Corn	15.8	GE (MJ/kg)	16.94
Soybean meal	28	DM%	89.62
Wheat bran	6.5	CP%	22.42
Corn gluten feed	4.5	EE%	3.07
DDGS	2.3	NDF%	52.42
Sunflower seed meal	4.6	ADF%	16.71
Expanded urea	0.5	Methionine	0.27
Soybean Oil	0.5		
Bone meal	0.6		
NaCl	0.7		
Premix^1^	1		
Corn stage	35		
Total	100		

ADF, acid detergent fiber; CP, crude protein; DM, dry matter; EE, ether extract; ME, metabolic energy; NDF, neutral detergent fiber. ME was a calculated value, while the others were measured values. Premix^1^: 1 kg of premix contained the following: MgO, 0.076 g; ZnSO_4_.H_2_O, 0.036 g; MnSO_4_.H_2_O, 0.043 g; FeSO_4_.H_2_O, 0.053 g; NaSeO_3_, 0.031 g; vitamin A, 2484 IU; vitamin D, 3496.8 IU; vitamin E, 0.828 IU; vitamin K, 0.23 mg; vitamin B1, 0.092 mg; vitamin B2, 0.69 mg; vitamin B12, 0.00138 mg; folic acid, 0.023 mg; nicotinic acid, 1.62 mg; calcium pantothenate, 1.15 mg; CaHPO_4_, 5.17 g; and CaCO_3_, 4.57 g.

**Table 2 animals-12-01950-t002:** Effect of methionine supplementation on productive performance and antioxidant and immune levels in sika deer.

Item	CON	LMet	HMet	*p*-Value
Antler weight (g)	1503.0 ± 381.7	1717.5 ± 132.7	1797.3 ± 183.4	0.29
Average daily gain of antler (g/d)	42.9 ± 10.9	49.1 ± 3.8	51.4 ± 5.2	0.29
IgA (ug/mL)	555.7 ± 13.1	580.7 ± 20.8	539.6 ± 54.2	0.28
IgG (mg/mL)	4.02 ^a^ ± 0.19	4.01 ^a^ ± 0.22	4.59 ^b^ ± 0.45	0.04
IgM (ug/mL)	103.7 ± 7.1	107.8 ± 4.1	114.3 ± 6.2	0.09
CAT (U/mL)	3.07 ± 0.76	2.76 ± 0.72	3.24 ± 0.34	0.58
GSH-PX (U/mL)	160.7 ^a^ ± 16.5	255.2 ^c^ ± 39.6	209.0 ^b^ ± 17.2	0.02
T-SOD (U/mL)	108.3 ± 10.0	104.6 ± 4.7	103.9 ± 9.4	0.74
T-AOC (U/mL)	0.90 ± 0.27	1.28 ± 0.43	1.65 ± 0.63	0.13

CAT, catalase; GSH-PX, glutathione peroxidase; IgA, immunoglobulin A; IgG, immunoglobulin G; IgM, immunoglobulin M; T-AOC, total antioxidant capacity; T-SOD, total superoxide dismutase. The values were expressed as mean ± standard deviation (SD). Means with different lowercase superscripts were significantly different at *p* < 0.05.

**Table 3 animals-12-01950-t003:** Effect of methionine supplementation on serum biochemical indexes in sika deer.

Item	CON	LMet	HMet	*p*-Value
TG (mmol/L)	0.14 ± 0.07	0.16 ± 0.09	0.16 ± 0.09	0.91
CHO (mmol/L)	1.96 ± 0.32	1.89 ± 0.19	1.89 ± 0.26	0.92
HDL-C (mmol/L)	1.58 ± 0.27	1.58 ± 0.04	1.66 ± 0.16	0.80
LDL-C (mmol/L)	0.29 ± 0.06	0.33 ± 0.03	0.30 ± 0.02	0.34
Glucose (mmol/L)	8.8 ± 1.6	7.4 ± 0.5	8.7 ± 1.3	0.21
TP (g/L)	60.9 ± 2.2	62.7 ± 4.2	60.4 ± 2.3	0.55
ALB (g/L)	22.1 ± 5.2	21.0 ± 3.9	21.3 ± 5.3	0.95
GLB (g/L)	38.8 ± 3.7	41.7 ± 2.9	39.1 ± 3.0	0.42
ALP (U/L)	451.9 ± 150.7	410.6 ± 122.6	402.3 ± 45.2	0.81
AST (U/L)	50.8 ± 7.6	51.7 ± 3.7	57.1 ± 13.1	0.59

ALB, albumin; ALP, alkaline phosphatase; AST, aspartate aminotransferase; CHO, cholesterol; GLB, globulin; HDL-C, high-density lipoprotein cholesterol; LDL-C, low-density lipoprotein cholesterol; TG, triglycerides; TP, total protein. The values were expressed as mean ± standard deviation (SD).

**Table 4 animals-12-01950-t004:** Effect of methionine supplementation on apparent amino acid digestibility in sika deer.

Item (nmol/mL)	CON	LMet	HMet	*p*-Value
Aspartic acid	81.1 ± 6.1	81.3 ± 4.3	80.9 ± 4.1	0.99
Threonine	76.0 ± 7.4	77.4 ± 7.1	75.8 ± 5.1	0.95
Serine	82.5 ± 5.5	83.3 ± 5.2	82.4 ± 4.0	0.97
Glutamic acid	87.2 ± 4.1	88.5 ± 4.4	87.2 ± 3.0	0.90
Glycine	77.8 ± 7.5	79.1 ± 6.7	77.6 ± 5.0	0.95
Alanine	79.9 ± 6.4	79.8 ± 4.0	79.4 ± 4.4	0.99
Cysteine	73.6 ± 8.2	77.6 ± 9.1	73.3 ± 5.5	0.76
Valine	80.6 ± 6.5	80.9 ± 4.1	80.5 ± 4.2	1.00
Methionine	74.2 ± 6.5	77.9 ± 9.4	72.5 ± 4.4	0.66
Isoleucine	81.1 ± 6.2	81.5 ± 4.3	81.1 ± 4.0	0.99
Leucine	86.5 ± 4.4	87.2 ± 3.1	86.5 ± 3.1	0.96
Tyrosine	82.5 ± 5.3	82.7 ± 3.9	82.2 ± 3.9	0.99
Phenylalanine	84.3 ± 5.2	84.5 ± 3.2	84.2 ± 3.5	1.00
Lysine	78.1 ± 7.0	77.9 ± 4.9	78.0 ± 4.6	1.00
NH3	83.5 ± 6.1	84.6 ± 2.6	84.1 ± 2.1	0.94
Histidine	85.8 ± 4.7	86.6 ± 3.6	86.2 ± 3.2	0.97
Arginine	87.8 ± 3.6	88.1 ± 2.8	87.8 ± 2.7	0.99
Proline	83.3 ± 5.4	83.6 ± 3.4	83.2 ± 3.9	0.99

**Table 5 animals-12-01950-t005:** Comparing the concentration of amino acids in the serum of sika deer among the three groups.

Item (nmol/mL)	CON	LMet	HMet	*p*-Value
Phosphoserine	0.36 ± 0.04	0.36 ± 0.04	0.38 ± 0.06	0.75
Taurine	3.2 ± 0.6	3.2 ± 0.6	3.1 ± 0.8	0.94
Urea	169.6 ± 6.9	188.6 ± 35.2	219.6 ± 50.7	0.19
Aspartic acid	0.59 ± 0.16	0.53 ± 0.04	0.57 ± 0.10	0.78
Threonine	1.5 ± 0.4	1.4 ± 0.2	1.3 ± 0.2	0.57
Serine	1.8 ± 0.3	2.0 ± 0.2	1.9 ± 0.3	0.49
Glutamate	2.2 ± 0.3	2.0 ± 0.2	2.2 ± 0.2	0.65
Alpha-aminoadipate	0.23 ± 0.11	0.21 ± 0.09	0.21 ± 0.09	0.94
Glycine	9.2 ± 0.9	8.5 ± 0.1	9.8 ± 1.4	0.21
Alanine	4.4 ± 0.5	4.8 ± 0.4	4.9 ± 0.3	0.35
Citrulline	2.3 ± 0.2	1.9 ± 0.2	2.1 ± 0.3	0.20
α-Aminobutyric acid	0.19 ± 0.04	0.17 ± 0.04	0.19 ± 0.08	0.78
Valine	5.9 ± 0.9	5.3 ± 0.6	5.7 ± 0.9	0.56
Methionine	0.87 ± 0.13	0.87 ± 0.11	0.90 ± 0.04	0.91
Cystathionine	0.24 ± 0.03	0.22 ± 0.02	0.24 ± 0.02	0.40
Isoleucine	1.9 ± 0.4	1.9 ± 0.3	1.8 ± 0.3	0.90
Leucine	3.2 ± 0.4	3.2 ± 0.5	3.4 ± 0.4	0.84
Tyrosine	1.0 ± 0.1	1.0 ± 0.2	1.1 ± 0.1	0.58
Phenylalanine	1.3 ± 0.1	1.3 ± 0.1	1.3 ± 0.1	0.43
Beta alanine	0.37 ± 0.05	0.37 ± 0.06	0.38 ± 0.06	0.96
NH3	3.4 ± 0.5	3.4 ± 0.5	3.8 ± 0.7	0.66
Hydroxylysine	1.03 ^a^ ± 0.15	0.83 ^b^ ± 0.05	0.74 ^b^ ± 0.08	0.01
Ornithine	2.1 ± 0.5	2.0 ± 0.3	2.3 ± 1.0	0.74
Lysine	2.3 ^a^ ± 0.4	2.1 ^ab^ ± 0.1	1.8 ^b^ ± 0.1	0.03
1-methylhistidine	0.49 ± 0.06	0.43 ± 0.04	0.41 ± 0.02	0.09
Histidine	1.3 ± 0.2	1.4 ± 0.1	1.3 ± 0.2	0.50
3-methylhistidine	2.0 ± 0.2	1.8 ± 0.2	2.0 ± 0.1	0.10
Carnosine	0.30 ± 0.05	0.28 ± 0.05	0.24 ± 0.07	0.37
Arginine	3.6 ± 0.8	3.8 ± 0.7	3.8 ± 0.7	0.94
Proline	2.0 ± 0.5	2.0 ± 0.2	2.2 ± 0.2	0.57

The values were expressed as mean ± standard deviation (SD). Means with different lowercase superscripts were significantly different at *p* < 0.05.

**Table 6 animals-12-01950-t006:** Effect of methionine supplementation on rumen fermentation.

Item	CON	LMet	HMet	*p*-Value
Acetate (mmol/L)	57.7 ± 4.0	61.9 ± 2.0	57.1 ± 6.5	0.63
Propionate (mmol/L)	17.0 ± 3.5	15.8 ± 3.2	15.7 ± 1.4	0.77
Isobutyrate (mmol/L)	1.3 ± 0.3	1.4 ± 0.3	1.2 ± 0.3	0.58
Butyrate (mmol/L)	6.9 ± 1.2	8.1 ± 2.3	6.7 ± 2.0	0.57
Isovalerate (mmol/L)	1.3 ± 0.4	1.6 ± 0.4	1.3 ± 0.4	0.52
Valerate (mmol/L)	0.45 ± 0.15	0.48 ± 0.13	0.44 ± 0.10	0.88
TVFAs (mmol/L)	84.7 ± 6.4	89.3 ± 16.7	82.3 ± 10.0	0.71
Ammonia (mg/dL)	11.1 ± 2.4	11.0 ± 2.6	10.7 ± 1.7	0.97
Microbial proteins (mg/mL)	1.8 ^a^ ± 0.2	2.6 ^b^ ± 0.4	2.1 ^ab^ ± 0.3	0.02
Rumen fluid pH	6.6 ± 0.1	6.8 ± 0.2	6.9 ± 0.2	0.07

TVFA, total volatile fatty acids. The values were expressed as mean ± standard deviation (SD). Means with different lowercase superscripts were significantly different at *p* < 0.05.

**Table 7 animals-12-01950-t007:** Adonis analysis of the bacterial communities in the rumen of sika deer.

Group	Unweighted UniFrac		Bray–Curtis		Weighted UniFrac	
	R^2^	*p*-Value	R^2^	*p*-Value	R^2^	*p*-Value
CON vs. LMet	0.33	0.03	0.29	0.02	0.47	0.02
CON vs. HMet	0.16	0.16	0.13	0.53	0.10	0.62
LMet vs. HMet	0.25	0.03	0.21	0.03	0.32	0.02

## Data Availability

The datasets generated in this study can be found in online repositories. The names of the repository/repositories and accession number(s) can be found below: https://www.ncbi.nlm.nih.gov/ (accessed on 25 June 2022), SRR198449519-SRR19849530.

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
