# Peer review of "Effect of Methionine Supplementation on Serum Metabolism and the Rumen Bacterial Community of Sika Deer (Cervus nippon)"

_animals, 2022, doi:10.3390/ani12151950_

Round 1
Reviewer 1 Report
Dear Authors
I read the manuscript with interest, and I attach below the comments that will help the authors to improve the presentation of the interesting results.
What were the criteria for using 4 deer for each group in the study?; seven is the minimum number of experimental units for these in vivo studies.
Why did they use only males?
I suggest adding or detailing this information in the manuscript.
The authors present the metabolic pathways of Glutathione peroxidase and the importance of measuring methionine; however, other thiol molecules have been reported to be involved in these processes.
I suggest that these molecules enter.
How does deer production in China contribute to economic development?
Include updated information about it.
The authors used the corn in the diets of the male deer, how did they ensure that it did not have glyphosate residues, to name a few?
In the experimental design section.
Because they used the DUNCAN comparison of means, does this mean that they did not standardize the variables before?, since DUNCAN is normally used for exploratory studies.
The suggestion is to use Tukey's comparison for robustness.
Add the data of the statistical package and the version.
Correct units such as “ml” to “mL” throughout the manuscript.
In the results section.
In table 2 and 3. Add the coefficient of variation data.
Attach the animal welfare letter for the use of deer.

Author Response
Reply to Reviewer 1
Deer Professor, thank you very much for your valuable suggestions on my manuscript, which made my content better. I will respond to your questions and suggestions in the following content.
1.
We don't have many experimental animals mainly because the value of the sika deer is relatively rare in China, especially the male sika deer, Because a male sika deer with excellent antler production can often be worth hundreds of thousands RMB, which is the key to the farm's economic benefits and prestige, and the farm owner is very worried about the damage to the deer due to the experiment, so it is very difficult to get more experimental animals. Four animals per group were also used in the previously published paper, and the prevailing number of animals in Chinese sika deer feed nutrition studies is 4-6, as can be seen in the following literature.
Huang J, Zhang T-T, Kun B, Li G-Y and Wang K-Y, Effect of Supplementation of Lysine and Methionine on Growth Performance, Nutrients Digestibility and Serum Biochemical Indices for Growing Sika Deer(Cervus Nippon)Fed Protein Deficient Diet. Italian Journal of Animal Science, 2015. 14(1): p. 3640.DOI: 10.4081/ijas.2015.3640.
Huang J, Sun W-L, Li C-Y, Liu H-L, Zhang T-T, Bao K, et al., Effects of DL-methionine supplement on growth performance and amino acid digestion and plasma concentrations in sika deer calves (Cervus nippon). Animal Production Science, 2016. 56(6): p. 1002.DOI: 10.1071/an15042.
Si H, Liu H, Nan W, Li G, Li Z and Lou Y, Effects of Arginine Supplementation on Serum Metabolites and the Rumen Bacterial Community of Sika Deer (Cervus nippon). Front Vet Sci, 2021. 8: p. 630686.DOI: 10.3389/fvets.2021.630686.
2.
Only males are used because only male deer grow antlers, females do not
3.
Thank you very much for your suggestion, we have described the metabolic process between methionine and glutathione in more detail in the manuscript, you can see the changes in the attachment
4.
At present, the number of antlered deer in China is around 700,000, and the value of the related industry is around 35 billion RMB, in addition to the meat deer and ornamental deer industries, but there is no clear data on the specific value. In 2020, China's Ministry of Agriculture and Rural Affairs officially announced the inclusion of the sika deer, red deer and reindeer in the National Breed List of Livestock and Poultry Genetic Resources, which means that deer farming entered into a management mode as important as common livestock such as pigs, cattle and sheep. In addition, this year, Jilin Province, China's largest deer farming province, issued industrial guidance, is expected to reach a total value of 100 billion RMB in the province's deer industry within five years. China's deer farming industry is ushering in development-based opportunities.
5.
Thank you very much for your concern about pesticide residues and environmental pollution, we are also aware of this problem. The feed we use is purchased from professional feed production companies who conduct strict quality tests before purchasing raw materials and selling products to ensure that there are no glyphosate residues in the feed or that the glyphosate residues meet international standards. At the same time, we also conducted the determination of glyphosate, aflatoxin and heavy metal content in the laboratory after we got the feed, and the check results met the requirements.
6.
Thank you very much for recommending the use of the Tukey test, which is a very rigorous test. However, for our experiment, this is the first exploration of methionine in the field of antler-growing stage sika deer feed nutrition, and at the same time, due to the difficulty of obtaining experimental animals, the number of experimental animals is not large enough, which can easily lead to the lack of significance of the experimental results, the Duncan test can more easily obtain the significant differences between groups, so that our experimental results have a good guide rather than being masked, in There are also many scholars in the field of sika deer feed nutrition using Duncan's test, you can refer to the literature.
Bao K, Wang K, Wang X, Zhang T, Liu H and Li G, Effects of dietary manganese supplementation on nutrient digestibility and production performance in male sika deer (Cervus Nippon). Anim Sci J, 2017. 88(3): p. 463-467.DOI: 10.1111/asj.12657.
Bao K, Wang X, Wang K, Yang Y and Li G, Effects of Dietary Supplementation with Selenium and Vitamin E on Growth Performance, Nutrient Apparent Digestibility and Blood Parameters in Female Sika Deer (Cervus nippon). Biol Trace Elem Res, 2020. 195(2): p. 454-460.DOI: 10.1007/s12011-019-01856-7.
7.
In the attachment we have updated the version information of the analysis software
8.
We are very sorry for the inconsistency of units in the manuscript, and we have made changes. Thank you very much for your guidance!
9.
The coefficient of variation data is available in the attached file
10.
Animal welfare information has been uploaded in the attachment, please check.
Changes have been made in conjunction with your suggestions and those of other reviewers, and you can view all changes in the attached document using the revision mode in word.
Since only one attachment can be uploaded in the review system, I added the attachment content of animal welfare information and coefficient of variation to the conclusion of the manuscript. After getting your approval, I will submit the attachment and manuscript that meet the format specification to the editor.
Thank you very much for your valuable suggestions on my manuscript, which brought to my attention a number of interesting issues
Best wishes!

Reviewer 2 Report
Ms. Ref. No.: animals-1813959
Title: “Effect of Methionine Supplementation on Serum Metabolism and the Rumen Bacterial Community of Sika Deer (Cervus nippon)”
Animals
General comments
I have had the opportunity to review the manuscript. The manuscript is interesting and is in the topic of the journal, however it needs more details in the methodology some points in the discussions
Below my considerations:
Introduction
The introduction should be expanded by making references to other species as well. The scope of the work needs to be more specifically detailed.
It is suggested that the following articles be read and added to also give environmental emphasis to this work:
Braghieri, A., Pacelli, C., Bragaglio, A., Sabia, E., Napolitano, F., 2015. The hidden costs of livestock environmental sustainability: The case of Podolian Cattle. In: Vastola, A. (Ed.), The Sustainability of Agro-food and Natural Resource Systems in the Mediterranean Basin, 2015, Springer Open, 47-56. https://doi.org/10.1007/978-3-319-16357-4_4
Sabia, E.; Napolitano, F.; Claps, S.; De Rosa, G.; Braghieri, A.; Pacelli, C. Dairy buffalo life cycle assessment as affected by heifer rearing system. J. Clean. Prod. 2018, 192, 647-655.
Serrapica F, Masucci F, Romano R, Napolitano F, Sabia E, Aiello A, Francia AD. Effects of Chickpea in Substitution of Soybean Meal on Milk Production, Blood Profile and Reproductive Response of Primiparous Buffaloes in Early Lactation. Animals (Basel). 2020 Mar 19;10(3):515.
doi: 10.3390/ani10030515
Material and methods
More details about the farm where the experiment took place should be provided: geographical coordinates, climatic conditions.
L 135-148 bibliography to be added
L 162-174 bibliography to be added
L 177-201 bibliography to be added
Conclusion
Conclusions must be expanded and extended, rewritten.
Author Response
Deer Professor, thank you very much for your valuable suggestions on my manuscript, which made my content better. Based on your suggestions, I have made changes, which are as follows.
Re:
Thank you very much for recommending these great articles to me, it makes me pay more attention to the problem of environmental pollution. We have revised the introduction to the manuscript to focus on the current state of environmental pollution in animal agriculture and cited your recommended article. You can view the specific changes in the uploaded attachment via the revision mode
We have added geographical coordinates and climatic conditions in the attachment and added references.
The experimental site is located in Shuangyang District, Jilin Province, China (Longitude: 125.724, Dimension: 43.539, Temperature: 19℃ - 28℃, Wind scale: <3).
3.
We have expanded on the conclusion. There is more discussion on the next step of thinking about the study
Changes have been made in conjunction with your suggestions and those of other reviewers, and you can view all changes in the attached document using the revision mode in word.
Finally, thank you again for your suggestions and help with my manuscript.
Best wishes!

Round 2
Reviewer 1 Report
Editor Dear
The authors addressed most of the comments; however, regarding the issue of glyphosate, they must show the results of the analyzes they carried out.
Best regards
Author Response
Deer Professor
Thank you very much for your patience and careful review of my manuscript and your careful attention to pesticide residues.
In response to your request, I have compiled the process and results of the previous feed glyphosate testing and uploaded them to the review system via attachment. Please review.
Thank you again for your help with my manuscript.
Best Wishes

Reviewer 2 Report
Accept in present form
Author Response
Dear Professor
Thank you again for your review and help with my manuscript. Thank you for your recognition of our research.
Best wishes